# Barriers and facilitators to reporting deaths following Ebola surveillance in Sierra Leone: implications for sustainable mortality surveillance based on an exploratory qualitative assessment

Mohamed F Jalloh  ,[1,2] John Kinsman,[1,3] James Conteh,[4] Reinhard Kaiser,[5] Amara Jambai,[6] Anna Mia Ekström,[1] Rebecca E Bunnell,[2] Helena Nordenstedt  [1]

► Prepublication history and supplemental material for this paper is available online. To view these files, please visit the journal online (http://dx.doi.org/10.1136/bmjopen-2020-042976).

For numbered affiliations see end of article.

**Correspondence to**
Dr Mohamed F Jalloh;
mohamed.jalloh@ki.se

## ABSTRACT

**Objectives** To understand the barriers contributing to the more than threefold decline in the number of deaths (of all causes) reported to a national toll free telephone line (1-1-7) after the 2014–2016 Ebola outbreak ended in Sierra Leone and explore opportunities for improving routine death reporting as part of a nationwide mortality surveillance system.

**Design** An exploratory qualitative assessment comprising 32 in-depth interviews (16 in Kenema district and 16 in Western Area). All interviews were audio-recorded, transcribed and analysed using qualitative content analysis to identify themes.

**Setting** Participants were selected from urban and rural communities in two districts that experienced varying levels of Ebola cases during the outbreak. All interviews were conducted in August 2017 in the post-Ebola-outbreak context in Sierra Leone when the Sierra Leone Ministry of Health and Sanitation was continuing to mandate reporting of all deaths.

**Participants** Family members of deceased persons whose deaths were not reported to the 1-1-7 system.

**Results** Death reporting barriers were driven by the lack of awareness to report all deaths, lack of services linked to reporting, negative experiences from the Ebola outbreak including prohibition of traditional burial rituals, perception that inevitable deaths do not need to be reported and situations where prompt burials may be needed. Facilitators of future willingness to report deaths were largely influenced by the perceived communicability and severity of the disease, unexplained circumstances of the death that need investigation and the potential to leverage existing death notification practices through local leaders.

**Conclusions** Social mobilisation and risk communication efforts are needed to help the public understand the importance and benefits of sustained and ongoing death reporting after an Ebola outbreak. Localised practices for informal death notification through community leaders could be integrated into the formal reporting system to capture community-based deaths that may otherwise be missed.

## Strengths and limitations of this study

► This large qualitative assessment helps to explain the complex reasons for the sharp and persistent decline in death reporting levels in Sierra Leone following the 2014–2016 Ebola outbreak.

► The assessment generated novel understanding of barriers and facilitators related to death reporting with themes that may be transferrable to other post-Ebola-outbreak contexts.

► Given that mortality surveillance is a key approach for identifying existing and new public health threats, the findings can help inform strategies for engaging community members to improve death reporting level.

► It is possible that some respondents may have provided socially desirable responses in terms of facilitators to report in order to match previously heard messages during the Ebola outbreak.

► Other key stakeholders with relevant views on mortality surveillance (eg, health workers and local officials) were not interviewed. Nevertheless, this assessment shed light on the perspectives of family members who failed to report the deaths of their loved ones as mandated by the government.

## INTRODUCTION

A popular traditional healer in a remote village in Kailahun district in Sierra Leone became ill and suddenly died around 30 April 2014 after treating patients from neighbouring Guinea.[1] The Sierra Leone Ministry of Health and Sanitation (MoHS) subsequently confirmed an outbreak of Ebola virus disease (Ebola) on 25 May 2014, which was linked to the burial of the traditional healer in Kailahun district.[2] Ebola cases quickly spread to neighbouring Kenema district, making the Eastern region the initial epicentre of the outbreak in Sierra Leone.[3] The epicentre of the outbreak shifted from the Eastern region

to the Western and Northern regions by September 2014. Over 14 000 people became infected in the country, of whom nearly 4000 died by the time the outbreak was declared over in November 2015 by the WHO.[2]

Traditional burials that involved various forms of physical contact with infected corpses were identified as core transmission amplifiers of the Ebola virus.[4] It has been estimated that nearly three new Ebola cases resulted from every unsafe traditional burial during the outbreak in West Africa.[4] Containing the outbreak required prompt identification of all deaths to ensure safe burial by teams trained on Ebola infection prevention and control.[5] Social mobilisation and risk communication efforts were implemented nationally and intensified in high-transmission districts to persuade communities to report all deaths to a national toll free telephone line using a short dialling code, 1-1-7.[6 7]

Mortality surveillance is a key approach for identifying and responding to public health threats[8] in both high-income countries[9] and low-income and middle-income countries[8 10–12] as part of routine surveillance[10 11] as well as in health emergency contexts including during disease outbreaks.[8 9 13] Mortality surveillance systems have been relied on to count the excess number of deaths due to an emerging health threat,[9] describe patterns in mortality occurrence[10 11] and help to quantify causes of death in a population.[12 14] Governments' vital registration systems have been used for national mortality surveillance purposes to monitor and describe deaths occurring in a country. In addition, or alternatively, sample-based mortality surveillance systems have been used to generate nationally representative data on deaths. Vital registration systems and sample-based systems have been combined into an integrated mortality surveillance system.[10] In other instances, mortality surveillance systems have focused on subpopulation groups (eg, children) within geographically defined subnational units.[12]

In Sierra Leone, death reporting through the 1-1-7 system constituted a critical component of identifying and responding to occurrences of deaths in communities to prevent unsafe traditional burials during the Ebola outbreak.[13] Previous analyses of monthly reports of call volumes to the 1-1-7 system showed a substantial decline across all districts after announcing the end of the outbreak.[13 15] In the last year of the outbreak, the average number of deaths reported monthly to the 1-1-7 system fell from approximately 9000 during January–October 2015, to 4000 during the enhanced surveillance period (November 2015–June 2016). The decline continued and reached less than 1000 reported deaths per month in 2017.[15] A national telephone survey was conducted in April 2017 to investigate the motivations of those who continued to report deaths to the 1-1-7 system after Ebola outbreak ended.[15] Results from the survey showed that people who reported deaths were more motivated to do so when Ebola-like symptoms were present in the deceased. However, since the survey only targeted individuals who had reported a death, barriers related to

reporting among those who failed to report to the 1-1-7 system were not understood. In this paper we examine death reporting barriers and explore opportunities for improving routine mortality surveillance in the aftermath of the Ebola outbreak in Sierra Leone.

## METHODS

In-depth qualitative interviews were conducted with 32 purposefully selected individuals who experienced a death in their household in April–August 2017 but did not report the death to the 1-1-7 system as required by the MoHS. Interviews were conducted by trained native speakers in two districts, one urban: Western Area (n=16) and one rural: Kenema district (n=16). Audio-recordings of the interviews were transcribed and the textual data were analysed using qualitative content analysis.[16] Consolidated criteria for Reporting Qualitative research guidelines are used to describe the methods of our qualitative exploratory assessment.[17]

### Setting

The initial purpose of the 1-1-7 system when it was established in 2013 was to get feedback from communities on the government's Free Health Care Initiative that provides essential primary healthcare services at no-cost to children under 5 years old, pregnant women and lactating mothers.[18] In August 2014, as Ebola cases began to surge, the government re-purposed the 1-1-7 line for communities to report suspected Ebola cases with a mandate that all deaths, even if not suspected of Ebola, must be reported for safe burial.[13] A national call centre was set up for triaging the call alerts to district personnel who dispatched ambulance and burial teams.[19] Alpren *et al* have comprehensively documented the implementation of the 1-1-7 system.[13]

The normalisation of death reporting in Sierra Leone during the Ebola outbreak presented a unique opportunity for the country to leverage the 1-1-7 system as a foundation for strengthening civil registration of vital statistics (CRVS) in the post-outbreak context. CRVS is part of global efforts to register all births and deaths occurring in all countries.[20] The 2015–2024 global strategic plan for CRVS aims to have functional country systems to record all deaths so that the WHO International Classification of Disease and Injuries can be used as 'the global standard for classifying causes of death in a comparable manner over time and between populations'.[20] Prior to the Ebola outbreak in Sierra Leone, deaths were supposed to be reported to the office of births and deaths through its local district offices as per legal mandate established in the 1983 Births and Deaths Registration Act.[21] The pre-Ebola-outbreak death registration level remains unknown. The paper-based CRVS death registration systems were never integrated with the digital platforms used in the 1-1-7 system during and after the Ebola outbreak.

The MoHS instituted enhanced surveillance mechanisms after the outbreak was declared over by WHO

because it anticipated possible flare-ups of new Ebola cases due to sexual transmission of the Ebola virus by male Ebola survivors, which had been reported in Liberia.[22 23] Also, the outbreak in Guinea had not been declared over by WHO, which meant that possible importation of cases needed to be monitored, especially along border regions.[24] Consequently, the MoHS mandated that all deaths must continue to be reported to the 1-1-7 system to help detect possible flare-up of new Ebola cases. Death investigations during the enhanced surveillance period included buccal swabbing of corpses to test for Ebola and determine the need for safe burial.[13] On 30 June 2016, the MoHS announced the end of enhanced Ebola surveillance.[13] The radio announcement stated that starting in July 2016 all deaths were still required to be reported through the 1-1-7 system, however, only deaths that were suspected of Ebola were required to be investigated by district-based surveillance teams. Buccal swabbing of corpses stopped after enhanced surveillance ended. The MoHS aimed to transition the 1-1-7 into the primary mechanism for death registration and mortality surveillance in Sierra Leone including for health workers to report facility-based deaths.

## Sampling

Western Area and Kenema districts were purposefully selected for inclusion in the exploratory assessment. Both districts were chosen because of their epidemiological significance during the 2014–2016 Ebola outbreak in Sierra Leone and to allow for rural–urban variations in the sample. Western Area was selected because it had the highest number of reported Ebola cases and it has a large urban population. Kenema was selected because it was an early epicentre of the outbreak and it has a large rural population. We expected that the sociodemographic characteristics and the different experiences of Ebola in these districts would facilitate in-depth understanding of the range of reporting barriers and possible facilitators of willingness to report in the future. Within each district, we conducted eight interviews in high Ebola burden communities and another eight in low burden communities. Ebola epidemiological data from the MoHS showing estimated case counts at subdistrict level guided our selection of communities. For this assessment, we defined high Ebola burden as ≥50 cumulative cases and low Ebola burden as ≤10 cases per community.

Trained data collection teams worked with community mobilisers to help identify households that had experienced one or more deaths between April and August 2017. The mobilisers contacted community leaders, including religious leaders, to inquire about deaths that occurred in the respective communities during the specified period. Based on the information gathered, the mobilisers and data collection teams visited the households referred by community leaders. For a referred household to be eligible to be included in the assessment, the death must not have been reported to the 1-1-7 system. We also used snowball sampling by asking eligible households to refer data collection teams to other households that may have experienced a death during the same period. All eligible households identified during recruitment agreed to participate in the assessment.

## Data collection

Data collectors then continued to follow-up with the identified households to confirm eligibility, explain the purpose of the assessment and ask for informed consent to participate. Once an eligible household was identified by the local team, informed consent was obtained from the head of the household or next of kin of the deceased. Only one interview was conducted per eligible household, and repeat interviews were not conducted. If the household head or next of kin were both unavailable, the data collection team returned to the household at least one more time before it was considered unreachable and another eligible household was approached. After obtaining informed consent from participants, the team used a structured questionnaire to gather basic demographic information about the respondent. This was followed by administering an in-depth interview using a semi-structured guide that covered two broad domains: community level practices and perceptions regarding the death; and personal experiences and perceptions regarding the death (online supplemental material). The interview guide was pilot tested with a convenience sample of four respondents as part of the training of the data collection teams. Feedback from the pilot was used to improve the framing and sequencing of questions and probes.

Data collection was carried out by mixed-gender teams of interviewers and note-takers with extensive prior experience in qualitative data collection in Sierra Leone. All data collection team members were fluent in the predominant local languages of their assigned districts (Krio in Western Area and Mende or Krio in Kenema), and had postsecondary education in social sciences or public health. All data collectors participated in a 1-week training that covered informed consent, sampling procedures, best practices for conducting interviews and debriefs and translation and transcription of audio-recordings. All interviews were audio-recorded with consent from respondents. On average the interviews lasted between 45 and 60 min. Interviews were conducted in a secluded area within the vicinity of the home. At the end of each interview, the interviewer and note-taker conducted a short debrief that lasted approximately 30 min in order to capture key topics that emerged from the discussion and to document important observations that may help to contextualise the responses. Review of the data and debrief notes indicated that analytical data saturation was achieved after 16 interviews were conducted in each district (ie, when meaningfully new information was no longer emerging from the interviews).

## Data management and analysis

The respective teams of interviewers and note-takers translated and transcribed the audio-recordings. Team members conducted peer reviews of each other's transcripts to ensure consistency in translations from local language to English. A supervisor reviewed all transcripts for translation accuracy and fidelity of meaning. However, interview transcripts were not provided to the participants for their review or correction. The analysis was led by the lead author (MFJ) with support from coauthor JC, both of whom are Sierra Leoneans with training and experience in qualitative data analysis. A web-based qualitative software, Dedoose,[25] was used to support the management and analysis of the data.

An initial set of deductive codes were first generated to reflect meaning units from the questions in the interview guide. The two analysts (MFJ and JC) generated additional inductive codes from reviewing the transcripts, then proceeded to code the transcripts, independently validated each other's application of codes and resolved any discrepancies. Textual excerpts were extracted from Dedoose for each code. The analysts iteratively reviewed, discussed and interpreted the coded excerpts. The final codes were organised into mutually exclusive subcategories and categories that reflected latent grouping of concepts. An iterative, interpretative process led to the higher-level grouping of the categories into themes. Preliminary results from the qualitative analysis were presented to stakeholders in Sierra Leone including the MoHS and other surveillance partners. Feedback received from the stakeholder presentation informed our interpretation and framing of the themes.

### Patient and public involvement

Patients and/or the public were not involved in the design, conduct, reporting or dissemination plans of this research.

## RESULTS

### Demographic characteristics of respondents

Out of 32 respondents, 18 (56%) were women, 15 (47%) had no education or only primary school education, 8 (25%) were petty traders and 22 (69%) self-identified as Muslims. Respondents' age ranged from 27 to 70 years; median age was 38 years. Respondents mostly comprised of relatives of deceased persons (29 out of 32), including their parents (n=12), spouses (n=5) and children (n=4) (table 1).

### Summary of themes

Death reporting barriers were driven by the lack of awareness to report all deaths, lack of services linked to reporting (eg, provision of ambulance services), negative experiences from the Ebola outbreak period including prohibition of traditional burial rituals, perception that inevitable deaths do not need to be reported and situations where prompt burials may be needed (table 2).

**Table 1** Sociodemographic characteristics of respondents, Sierra Leone, 2017

| Characteristic | Number of respondents (N=32) |
| --- | --- |
| Sex | |
| Female | 18 |
| Male | 14 |
| Education | |
| None or primary only | 15 |
| Secondary or higher | 17 |
| Religion | |
| Muslim | 22 |
| Christian | 10 |
| Occupation | |
| Petty trader | 8 |
| Skilled labour | 7 |
| Private business | 4 |
| Teacher | 3 |
| Student | 3 |
| Unemployed | 3 |
| Driver/bike rider | 2 |
| Farmer | 1 |
| Civil service | 1 |
| Age (years) | Median=38 |
| 21–30 | 6 |
| 31–40 | 15 |
| 41–50 | 4 |
| 51–60 | 6 |
| 61–70 | 1 |
| Relationship to the deceased person | |
| Parent | 12 |
| Spouse | 5 |
| Child | 4 |
| Grand parent | 3 |
| Non-relative | 3 |
| Sibling | 3 |
| Other relative | 2 |

Facilitators of future willingness to report deaths were largely influenced by the perceived communicability and severity of the disease, unexplained circumstances of the death that need investigation and the potential to leverage existing death notification practices through local leaders (table 3). We did not observe any substantive differences in the thematic findings between the two districts and areas within districts.

### Barrier 1.1: lack of awareness to report all deaths

All respondents were unaware that they were required to report the household death after the end of the enhanced Ebola surveillance. Although some respondents knew

**Table 2** Thematic area on barriers to reporting deaths in the aftermath of the Ebola outbreak, Sierra Leone, 2017

| Code | Category | Theme |
|---|---|---|
| 1.1.1 Reporting only required during Ebola | 1.1 Lack of awareness to report all deaths | Barriers to reporting deaths in the aftermath of the Ebola outbreak |
| 1.1.2 No more need to report to 1-1-7 | | |
| 1.1.3 Only Ebola-like deaths should be reported | | |
| 1.2.1 No services for people while alive | Lack of reciprocal benefit | |
| 1.2.2 Too much focus on dead people | | |
| 1.2.3 Nothing done for sick people | | |
| 1.2.4 Nothing happens if you report | | |
| 1.2.5 No help with burial | | |
| 1.2.6 Just a government line | | |
| 1.3.1 Old-age/God's time | 1.3 Perceived inevitability of certain deaths | |
| 1.3.2 Long-term illness or disability | | |
| 1.4.1 Islamic requirement to bury within 24 hours | 1.4 Needing to bury promptly | |
| 1.4.2 Body recovered from drowning | | |
| 1.4.3 Body recovered from fire | | |
| 1.4.4 Young child | | |
| 1.4.5 Sick for long time | | |
| 1.5.1 Wanting to forget about 1-1-7 | 1.5 Negative experiences from the Ebola outbreak | |
| 1.5.2 Reporting brings sadness to family | | |
| 1.5.3 Painful memory of Ebola | | |
| 1.5.4 Burial delays during Ebola | | |
| 1.5.5 Restrictions on traditional burials | | |
| 1.5.6 Fear of quarantine | | |
| 1.5.7 Fear of ambulance | | |
| 1.5.8 Line was used during Ebola | | |
| 1.5.9 Do not like the number | | |

**Table 3** Thematic area on facilitators of future intention to report deaths, Sierra Leone, 2017

| Code | Category | Theme |
|---|---|---|
| 1.1.1 Resembling Ebola | 1.1 Presence of Ebola-like symptoms | Facilitators of willingness to report deaths in the future |
| 1.1.2 Resembling Lassa | | |
| 1.1.3 Bleeding before dying | | |
| 1.2.1 Sudden death without illness | 1.2 Sudden and unexplained death | |
| 1.2.2 Wanting to know cause of death | | |
| 1.3.1 Informing chief | 1.3 Existence of other local reporting mechanisms | |
| 1.3.2 Informing religious leaders | | |
| 1.3.3 Informing elders | | |
| 1.3.4 Inform city council for burial permit | | |
| 1.3.5 Informing office of births and deaths for death certificate | | |

that the 1-1-7 line was still operational, they thought that only deaths that resembled Ebola needed to be reported.

> … I do not think 1-1-7 is still existing because after Ebola we thought that was the end of 1-1-7. I'm only hearing this from you now. I always listen to the radio, but it has taken a long time I did not hear announcement that when someone dies, we are to call 1-1-7; even in the villages, that is why I did not remember to call 117. – respondent from Western Area district

**Barrier 1.2 lack of reciprocal benefit**

Respondents did not see a benefit to report deaths to the 1-1-7 system in a post-Ebola-outbreak setting. The notion of simply notifying 1-1-7 without any associated follow-up action or service was not appreciated by interviewees. To report deaths to 1-1-7, respondents said they would expect some follow-up action or service to be provided. For instance, they strongly recommended for prompt ambulance services for sick people and transportation of corpses through 1-1-7. Respondents expressed that such services would help motivate them in the future to use the system.

> …if you take transport, like you take a taxi, to carry a [sick] person to the hospital; when you are going with him and there is traffic, they won't give you way. But let's say you call the 1-1-7 and the 1-1-7 comes with the ambulance, they will be able to give you way because they will take it as an emergency. – respondent from Western Area district

> Then I would like them to give us ambulance in the community so when someone dies, they will be able to take the person and bury him/her quickly. – respondent from Kenema district

**Barrier 1.3 perceived inevitability of certain deaths**

Respondents consistently expressed that they would not consider reporting deaths that they perceive to be inevitable due to old age, God's will and long-term illness or disability.

> The illness that affects someone for so long, for example stroke [complications], which leads to death, we will not report such death. For instance, in our community, we had a man by the name of xxx [redacted] who was affected by stroke and had suffered with it for a very long time; his family had tried all forms of medication, but he did not survive. So, with this type of death the chief themselves will just give permission to the people for burial rather than reporting to 1-1-7. – respondent from Kenema district

### Barrier 1.4 needing to bury promptly

There were two main reasons why prompt burial emerged as a barrier to death reporting. First, circumstances of the death influenced perceptions of when the corpse should be buried including the death of young children, someone who has been ill for a long time (despite their age) and someone who died from an accident (eg, drowning, fire, road accident). Second, Muslim respondents emphasised that they need to bury the corpse within 24 hours to honour Islamic requirements; they feared that reporting may result in burial delay based on their experiences from the Ebola outbreak. One respondent gave an example of how same-day burial was done as per Islamic tradition:

> I went straight to the Imams [at the local mosque] and notified them [of the death]. The Imam came and asked us to take the corpse to the Mosque. The corpse laid there until around twelve o'clock when they washed and wrapped it with Kasankay [white] cloth. We waited for the time … that is two o'clock, then we went to bury him. After the burial the 'Jamat' [mosque congregation] met here [at our house], ate and prayed for him before everyone went back to their homes. – respondent from Western Area district

### Barrier 1.5 negative experiences from the Ebola outbreak

The 1-1-7 system was intricately linked to its widespread use during the Ebola outbreak in Sierra Leone and could not be separated by respondents' negative experiences with how some deaths were handled by burial teams during the outbreak.

> The family would not get access to the corpse or even go close to it. So this knowledge had existed within people that 1-1-7 is not a call to make in order to get help in the burial of their loved ones. They will only come to do whatever they feel like, whether it is in a respectful way or not; they don't care. So, with that, people in the community do not favour the 1-1-7. Because how we expect them to bury our loved ones it's the opposite [that they will do]. – respondent from Kenema district

Most participants cited that burial methods used to bury their loved ones during the Ebola outbreak discouraged them from reporting to 1-1-7.

> This 1-1-7 line…I don't want it. I want us to be respecting the [dead] people because the
> 1-1-7 was not burying our people properly. So, we are burying our people. Let government leave it [burial] up to us. If a doctor checks the body [that's fine], but don't let the 1-1-7 - come here until we have buried the corpse. – respondent from Kenema district

Dissatisfaction with burial methods was coupled with discontent about delays by burial teams when they were responding to death alerts during the Ebola outbreak.

> Like during the Ebola time if you call…they [burial teams] will not come [on time]. They will abandon them for some time before coming to take the person. All they care about is for the people to call them… They need to put more efforts into how they respond and treat the people with respect. – respondent from Western Area district

Fear of being quarantined was also mentioned as a barrier for reporting deaths to 1-1-7 as well as the sounds made by the ambulances and spraying of the house with chlorine, which were all associated with Ebola-related stigma. Respondents expressed that just the thought of 1-1-7 alone would usually bring back bitter memories of Ebola.

> The first time I heard about 1-1-7 was during Ebola and each time I hear about 1-1-7 I think about Ebola at once. The moment they talk about 1-1-7, it's a worry for me because during that period, when people see 1-1-7 coming, everybody would run away. When they come to a place, they will spray chlorine all over and everybody avoided body contact like nobody's business, and that worried us too much. That is why we hardly forget about 1-1-7. – respondent from Western Area district

### Facilitator 1.1 presence of Ebola-like symptoms

Knowledge gained during the Ebola outbreak influenced respondents' perceptions of the deaths that should be reported to the 1-1-7 system after the outbreak ended. Participants expressed that they would report a death if it resembled Ebola or Lassa fever, especially in situations where the person bled before dying.

> For any death pertaining to what government told us [we need to report]. That of a bad disease like Ebola, Lassa fever…The people around will not even dare to touch the person, because it is a transferable disease and it is more common in Eastern Province. – respondent from Kenema district

### Facilitator 1.2: sudden and unexplained deaths

Sudden and unexplained deaths wherein the person was not previously sick were perceived as needing to be reported to the authorities for further investigations.

> Like I said before, when someone dies abruptly, and nothing was wrong with him [before dying]. I will just be looking at him, I will not have the knowledge to know the cause of death, I will not have the machine to show that this is the sickness that caused the death or whether he just fell down and died or whether he just sat down and died. When you go to a medical person [through 1-1-7], he can confirm that this is the cause of the death. If the doctor has confirmed that for real he has died, what can I do? – respondent from Western Area district

### Facilitator 1.3: existence of other local reporting mechanisms

Informing local leaders—such as religious leader, chiefs and village headmen—about the death was a common practice that most respondents mentioned. In Kenema, some participants cited that informing local and traditional heads granted them permission for burial without needing to report to the 1-1-7 system. The reporting of these deaths to only local leaders showed that respondents were generally willing to report the deaths but only did so in localised ways outside of the 1-1-7 system.

> Well you will have to go and say to the chief or authority that someone has died amongst us. The chief will ask what happened to that person. The chief will ask you… and you will say [something like] it was a cold, or after two three days I noticed that this person was ill. God has taken his life. This is the way he died. The chief will ask you; and you should answer. You have the chance to report to the police station. You have the chance to report to the chief. And you have the chance to call the family members. – respondent from Kenema district

### DISCUSSION

In the aftermath of the Ebola outbreak in Sierra Leone, we identified barriers that prevented people from reporting deaths to the authorities, and we also explored facilitators that would encourage death reporting as part of routine mortality surveillance. Barriers uncovered in our assessment were linked to a lack of awareness to continue reporting all deaths after enhanced Ebola surveillance ended as well as the lack of reporting benefits. Respondents were unaware of the requirement to continue reporting all deaths after the enhanced surveillance period, and they were under the impression that only deaths resembling Ebola should be reported to the 1-1-7 system. Consequently, other deaths that were not suspected of Ebola were not reported to the 1-1-7 system. A separate assessment found that after the outbreak ended, people were more motivated to report deaths to the 1-1-7 system if Ebola-like symptoms were present in the decedent.[15] Our findings further demonstrated that although respondents did not report deaths through the 1-1-7 system as mandated by their government, they informed local councils and local leaders about the deaths. Therefore, integrating localised practices for death reporting into routine surveillance mortality systems may help optimise the number of deaths captured. Respondents reported about the lack of any reporting benefits associated with death reporting in a post-Ebola-outbreak context. They wanted ambulance services to be linked to 1-1-7 reporting as done during the outbreak. Contextually, past experiences from the Ebola outbreak served as both facilitators and barriers. Past outbreak experiences reinforced the importance of reporting when Ebola is suspected to avoid transmission risks. On the other hand, past experiences that involved dissatisfaction with how burials were handled during the outbreak may have discouraged reporting to the 1-1-7 system after the outbreak ended.

Despite efforts to promote reporting of all deaths during the outbreak, communities continued to express dissatisfaction with how their loved ones were buried and there were instances of secret burials that occurred outside of the safe burial process, which may have been due to discontent with safe burials or wanting to comply with secret society practices, for instance.[26 27] Dissatisfaction with the burial process derailed community trust to report deaths. Although dissatisfaction persisted regarding Ebola safe burial processes, communities were willing to comply with reporting because they wanted to help end Ebola transmission in the country.[28] In the waning period of the Ebola outbreak in 2015, an assessment of the community event-based surveillance showed that over 12 000 reports were submitted and investigated, out of which 287 met case-definition for suspected Ebola and 16 were confirmed positive for Ebola.[29] In that assessment, it was revealed that community event-based surveillance detected four new Ebola cases that were not epidemiologically linked and could have gone undetected.

The Ebola outbreak was tragic in many ways as demonstrated by the thousands of lives lost and the unquantifiable suffering inflicted on the people of Sierra Leone. The pain and misery they endured was evident in our assessment when they talked about their experiences during the outbreak. Even though people were sometimes dissatisfied with how the burials were handled or delayed, they recognised that it was to their benefit to have a safe burial to avoid household transmission risk. Nevertheless, normalising death reporting during the outbreak required gaining the trust of communities by engaging them to appreciate the benefits of reporting. The halting of ambulance services after the Ebola outbreak ended prompted people to question why those services were only provided during the outbreak response.

The principle of positive reciprocity has been well established in social psychology,[30–32] which implies that people become motivated to comply with a request when they receive something in return for their action. This notion is also supported in the health behaviour literature. For instance, applications of the health belief model have shown that the perceived benefit associated with a behaviour is a strong predictor of engaging in the behaviour.[33 34] Our findings are consistent with notions of reciprocity; respondents expressed that they will be willing to report deaths if tangible benefits are provided in return for complying with reporting.

Although it took time to establish trust between the government and communities to achieve high level of death reporting during the Ebola outbreak,[13 35 36] communities had eventually come to expect certain services in return after reporting a death (transportation to the burial ground and laboratory testing of the corpse) and information (communication of laboratory results to the family).[13 28 35] Although similar services may not be feasible or applicable in the routine mortality surveillance

environment, there is an opportunity at the community level to provide aggregated information about the deaths back to the community (eg, through community leaders and community-based organisations) as a form of reciprocal action to foster dialogue on addressing community level health threats.[37] Another key finding from our assessment is that people want help for sick family members who are still alive. Linking the country's expanding fleet of 170 ambulances[38] with the 1-1-7 toll free line could help promote a feeling of reciprocity in addressing other health needs in the community for people experiencing health emergencies.

## Limitations

The exploratory qualitative assessment is subject to several limitations. The results are not generalisable beyond the 32 individuals interviewed. While this is an inherent limitation to most qualitative research, it is important to note that we never intended to produce generalisable results. Instead, our aim was to generate in-depth understanding of barriers and facilitators of death reporting with themes that may be transferrable to other local contexts. It is possible that some respondents may have provided socially desirable responses in terms of facilitators to report to match previously heard messages during the Ebola outbreak. Because theoretical sampling was not used, as done in grounded theory approaches, for example,[39] other individuals outside of family members (eg, health workers, local city council officials) were not interviewed. Additional research with more diverse stakeholders may help shed light on additional barriers and facilitators.

## Conclusion

Respondents misunderstood the policy of reporting all deaths after the end of enhanced Ebola surveillance in Sierra Leone, which may have been due to communication gaps in the government's official announcement of the reporting mandate. We found that respondents perceived that only suspected Ebola deaths should be reported to the 1-1-7 system. The lack of awareness to report all deaths and the lack of perceived reporting benefits were the main reasons for failure to report the deaths. The post-outbreak death reporting policy should consider integrating community level benefits to encourage reporting. Existing practices for informal death notification through local leaders should also be leveraged to capture community-based deaths that may be missed by the formal reporting system. For example, establishing a reporting mechanism through trusted local religious and traditional leaders could help to strengthen reporting levels since these leaders are almost immediately notified of deaths that occur in their communities. Improving routine death reporting may be supported by well-planned social mobilisation efforts to educate communities about the death reporting policy, promote the reporting benefits and facilitate optimal compliance.

**Author affiliations**
[1]Department of Global Public Health, Karolinska Institutet, Stockholm, Sweden
[2]Division of Global Health Protection, U.S. Centers for Disease Control and Prevention, Atlanta, Georgia, USA
[3]Department of Epidemiology and Global Health, Umeå University, Umeå, Sweden
[4]Independent Consultant, Freetown, Sierra Leone
[5]Sierra Leone Country Office, U.S. Centers for Disease Control and Prevention, Freetown, Sierra Leone
[6]Sierra Leone Ministry of Health and Sanitation, Freetown, Sierra Leone

**Acknowledgements** We thank the 32 participants for the invaluable information they provided us under difficult circumstances. This paper is dedicated to the memories of their loved ones.

**Contributors** MFJ led the conceptualisation and design of the assessment with primary support from JK, RK and HN and secondary support from AJ, AME and REB. MFJ trained and supervised data collectors. MFJ analysed the data with support from JC and guidance from JK and HN. MFJ, JK, JC, RK, AJ, AME, REB and HN contributed to the interpretation of the results. MFJ led the writing of the manuscript with contributions from JK, JC, RK, AJ, AME, REB and HN. All authors reviewed and approved the final version of the manuscript.

**Funding** Data collection for this work was supported by US Centers for Disease Control and Prevention through a cooperative agreement with eHealth Africa.

**Disclaimer** The findings and conclusions in this report are those of the authors and do not necessarily represent the official position of the US Centers for Disease Control and Prevention, Karolinska Institutet, eHealth Africa or Sierra Leone Ministry of Health and Sanitation.

**Competing interests** None declared.

**Patient consent for publication** Not required.

**Ethics approval** The assessment was approved by the Sierra Leone Ministry of Health and Sanitation as a routine public health activity. In addition, the Center for Global Health at the US Centers for Disease Control and Prevention approved the assessment as a programme evaluation activity (CGH HSR Tracking # 2017-327).

**Provenance and peer review** Not commissioned; externally peer reviewed.

**Data availability statement** All data relevant to the study are included in the article or uploaded as supplemental information.

**ORCID iDs**
Mohamed F Jalloh http://orcid.org/0000-0002-7206-8042
Helena Nordenstedt http://orcid.org/0000-0002-9226-6441

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
