## [Reviewer comments · BMJ Open]

ARTICLE DETAILS

TITLE (PROVISIONAL)	Barriers and facilitators to death reporting following Ebola surveillance in Sierra Leone: Implications for sustainable mortality surveillance
AUTHORS	Jalloh, Mohamed; Kinsman, J; Conteh, James; Kaiser, Reinhard; Jambai, Amara; Ekström, Anna; Bunnell, Rebecca; Nordenstedt, Helena

VERSION 1 – REVIEW

REVIEWER	Adokiya, Martin University for Development Studies, Global and International Health
REVIEW RETURNED	09-Nov-2020

GENERAL COMMENTS	Comments to the Authors: General Comments: The topic addresses an important issue in health particularly in development countries. Mortality records are incomplete and unreliable. This study contributes to the overall health system strengthening in many countries. The topic is clear. However, there are abbreviations under key words. The authors should write the abbreviations in full. Title: The title is very clear. Abstract: 1. Objectives: The objective is well stated. However, it is unclear to verify the expression "sharp decline in reporting" (page 2, lines 6-7). This makes the sentence somehow vague.2. Design: The design is appropriate for this particular study. However, it has some limitations.3. Setting: The authors indicated that 32 indepth interviews were conducted from two districts. It may be helpful to state the number of respondents per districts. This important because the districts had varying levels of Ebola cases. In addition, the word "in" has been repeated (page 2, line 21).4. Participants: The participants were deceased members.5. Results: The results covering barrriers and facilitators are clearly stated in the abstract.6. Conclusion: The first sentence of the conclusion does not reflect the results (page 2, lines 42 -43). That is, the need
---

	to incentivize for death reporting. Some participants wondered why the ambulances were no longer available post Ebola outbreak. However, it is not clear if that will improve death reporting. Main Manuscript: Introduction:  1. The sentence "... confirmed an outbreak of Ebola Virus Disease (Ebola) .." (page 4, line 9). This should be revised to read as "...Ebola Virus Disease (EVD)..". 2. The sentence with the word "Ebolavirus" should be changed to "Ebola virus". It is expressed as two words (page 4, line 21). 3. The sentence "... but also in other in post public health emergency settings" (page 5, line 9). This sentence is unclear. It needs revision. 4. The sentence, "However, Interview..." the word interview started with Upper Case. It should be changed to lower case (page 8, line 14). Results:  1. The sentence "So, with this type of death the chief themselves...". This needs revision (page 10, line 19). Conclusion: The conclusion is clear. The authors may reduce the length to be concise. In addition, the linked between ambulances and possibility of improving death reports should not be removed.
--	---

REVIEWER	Raven, Joanna Liverpool School of Tropical Medicine, Department of International Public Health
REVIEW RETURNED	26-Nov-2020

GENERAL COMMENTS	This is an interesting paper on an important topic. It is well written. However I have a few concerns:  1. A brief section on the importance of mortality surveillance and why this is important in all settings and in LMIC is needed. It would be good to include more learning from other settings about how to engage with communities about routine death reporting. 2. Ethics:  • These interviews involve talking with people who have experienced a death of a relative including children, as well as asking people about their experiences during Ebola outbreak. This is highly sensitive and has the potential for causing distress. How did the research team prepare and handle this? • Ethics approval: it does not appear that ethical approval was sought or received but rather approved as a routine public health activity, and a program evaluation activity. Please could you explain more about your ethical conduct of this study as there are many ethical issues associated with this study.
--

	3. Methods:  • the design does not include family members who did report deaths during this period. It would be good to explain why these were not included as they could provide valuable insights; and include in the limitations. • It is unclear why only 66% of respondents were family members of the deceased (as this sampling criteria were household members); who were the other 34%? • It is unclear when these interviews were conducted and they refer to deaths that happen in 2017. Is there a significant lag in time between the death and the interview – what are the implications of this in terms of the perceptions and experiences of the participants? 4. Results:  • In the results summary – it is unclear what you mean by lack of services linked to reporting – needs to be more clearly aligned with the barrier 1.2 described in page 9. • Were there differences in perceptions and experiences of people living in the 2 districts and the areas within the districts and by gender (as outlined in the sampling section). This has not come across in the results or the discussion. • Perceived inevitability of certain deaths barrier: the quote does not seem to fit in this barrier as it indicates that permission for burials is given by the chief – shows the importance of existing community structures and practices. In addition, more explanation of how people perceived deaths from minor illness, drowning or fire as being inevitable and not being reported. 5. Discussion:  • this needs to be more aligned with the results emerging from this study. • if there were differences between the districts these should be discussed. • The positive reciprocity section needs to be more related to the findings, and how this can help with mortality reporting going forward. • There is a suggestion on integrating localised practices for death reporting into routine surveillance systems – can you elaborate how this could be done. • Including ambulance services as a benefit – is this realistic given the limited resources within the health system in Sierra Leone? • Mortality surveillance is one aspect that illustrates the level of trust between communities and the government and health system – this should be discussed more, and how this can be improved. Trust in the health system, and reporting system is clearly important but this is not really discussed in the discussion – how do you develop that trust, how is this data used, what are the benefits to the community and how can these be communicated effectively.
--	---

Reviewer 1. Dr Martin Adokiya, University for Development Studies

General Comments:

The topic addresses an important issue in health particularly in development countries. Mortality records are incomplete and unreliable. This study contributes to the overall health system strengthening in many countries.

RESPONSE: We very much appreciate your positive appraisal of our manuscript and thank you for your feedback.

The topic is clear. However, there are abbreviations under key words. The authors should write the abbreviations in full.

RESPONSE: Thank you for raising this to our attention. We have written out all abbreviations.

Title:

The title is very clear.

RESPONSE: Thank you.

Abstract:

1. Objectives: The objective is well stated. However, it is unclear to verify the expression "sharp decline in reporting" (page 2, lines 6-7). This makes the sentence somehow vague.

RESPONSE: We have quantified the extent of the decline by specifying the following: "...more than three-fold decline in the number of deaths..."

2. Design: The design is appropriate for this particular study. However, it has some limitations.

3. Setting: The authors indicated that 32 in-depth interviews were conducted from two districts. It may be helpful to state the number of respondents per districts. This is important because the districts had varying levels of Ebola cases. In addition, the word "in" has been repeated (page 2, line 21).

RESPONSE: We have specified in the abstract that "32 in-depth interviews (16 in Kenema district and 16 in Western Area)."

4. Participants: The participants were deceased members.

RESPONSE: We want to clarify that we interviewed family members of deceased individuals.

5. Results: The results covering barriers and facilitators are clearly stated in the abstract.

RESPONSE: Thank you.

6. Conclusion: The first sentence of the conclusion does not reflect the results (page 2, lines 42 -43). That is, the need to incentivize for death reporting. Some participants wondered why the ambulances were no longer available post Ebola outbreak. However, it is not clear if that will improve death reporting.

RESPONSE: We have removed the first sentence, which also keeps the word count the

300-word limit set by the journal (given the new additions to the abstract).

Main Manuscript:

Introduction:

1. The sentence "... confirmed an outbreak of Ebola Virus Disease (Ebola) .." (page 4, line 9). This should be revised to read as "...Ebola Virus Disease (EVD).."

RESPONSE: We appreciate the suggestion. However, we prefer to refer to the disease as 'Ebola' as opposed to EVD. We defer to the BMJ Open's editorial team regarding the journal's preference.

2. The sentence with the word "Ebolavirus" should be changed to "Ebola virus". It is expressed as two words (page 4, line 21).

RESPONSE: In this instance we are referring to the pathogen not the disease; hence why we have specified as one word. Similar to above, we defer to the BMJ Open's editorial team regarding the journal's preference.

3. The sentence "... but also in other in post public health emergency settings" (page 5, line 9). This sentence is unclear. It needs revision.

RESPONSE: We have removed the sentence.

4. The sentence, "However, Interview..." the word interview started with Upper Case. It should be changed to lower case (page 8, line 14).

RESPONSE: It has been changed to lowercase.

Results:

1. The sentence "So, with this type of death the chief themselves..." This needs revision (page 10, line 19).

RESPONSE: This is a verbatim quote from a participant that was locally translated. To ensure authenticity, we have opted to keep the quote how it was translated by the local team.

Conclusion:

The conclusion is clear. The authors may reduce the length to be concise. In addition, the linked between ambulances and possibility of improving death reports should not be removed.

RESPONSE: We have shortened the conclusion as per the feedback.

Reviewer 2. Dr. Joanna Raven, Liverpool School of Tropical Medicine

Comments to the Author:

This is an interesting paper on an important topic. It is well written. However, I have a few concerns.

RESPONSE: Thank you for the feedback, we have provided point-by-point responses in attempt to address each concern you have raised.

1. A brief section on the importance of mortality surveillance and why this is important in all settings and in LMIC is needed. It would be good to include more learning from other settings about how to engage with communities about routine death reporting.

RESPONSE: We have expanded on the third paragraph in the introduction to summarize the role of mortality surveillance across diverse contexts and settings:

“Mortality surveillance is a key approach for identifying and responding to public health threats⁸ in both high income countries⁹ and low- and middle-income countries^{8 10-12} as part of routine surveillance^{10 11} as well as in health emergency contexts including during disease outbreaks.^{8 9 13} Mortality surveillance systems have been relied upon to count the excess number of deaths due to an emerging health threat,⁹ describe patterns in mortality occurrence,^{10 11} and help to quantify causes of death in a population.^{12 14} Governments’ vital registration systems have been used for national mortality surveillance purposes to monitor and describe deaths occurring in a country. In addition, or alternatively, sample-based mortality surveillance systems have been used to generate nationally representative data on deaths. Vital registration systems and sample-based systems have been combined into an integrated mortality surveillance system.¹⁰ In other instances, mortality surveillance systems have focused on sub-population groups (e.g. children) within geographically defined sub-national units.^{12”}

2. Ethics:

- These interviews involve talking with people who have experienced a death of a relative including children, as well as asking people about their experiences during Ebola outbreak. This is highly sensitive and has the potential for causing distress. How did the research team prepare and handle this?

RESPONSE: As the reviewer has rightfully pointed out, there were unique ethical dimensions in this assessment that required careful consideration and planning. Interviews were conducted with individuals who had recently experienced the death of a loved one – usually a close family member. As such we anticipated a range of related issues that we incorporated into the training of the data collectors, recruitment processes, administration and supervision of interviews, management of the data, data analysis and data reporting. We worked with local leaders and mobilizers during the recruitment phase who helped introduce our team to the families, which helped in building rapport and establishing some baseline of trust ahead of the interview. Data collectors were trained to sensitively approach potential eligible participants by working with local leaders and community mobilizers from the communities. We intentionally only approached potential participants whose loved ones died at least a month ago to help ensure that some time would have elapsed to allow for some period of grieving before the interview. We trained data collection teams on best practices for rapport-building and informed consent procedures that greatly emphasized voluntary participation and the ability for participants to end the interview at any given time for any reason. Interviewers were also trained and given guidance to halt the interview if a participant appeared to be in emotional distress during any time of the interview; and to document the situation and notify the supervisor immediately.

Building on our prior assessment focusing on individuals who reported deaths in the post-Ebola epidemic period to the national 1-1-7 call center (<https://pubmed.ncbi.nlm.nih.gov/32810138>), we gained practical experience interviewing a large sample of more than 1000 survey respondents who experienced a death in their household, community, or health facility. We applied lessons learned from that experience to train data collection teams on how to ask questions sensitively and respectfully in ways that minimized evoking negative emotional reactions among respondents. During the training, we developed and explored various scenarios of emotional reactions from participants and worked with the local team to develop

culturally appropriate ways to address them. In addition, we had a dedicated team of supervisors who rotated to observe the interviews and gave feedback to interviewers. Supervisors debriefed with teams to get feedback on each interview, discuss issues/concerns, and identify appropriate mitigation steps for subsequent interviews where applicable. Careful steps were taken to ensure that all data were de-identified, and that anonymity was always maintained in the reporting of the data.

- Ethics approval: it does not appear that ethical approval was sought or received but rather approved as a routine public health activity, and a program evaluation activity. Please could you explain more about your ethical conduct of this study as there are many ethical issues associated with this study.

RESPONSE: We want to clarify that within CDC, when any data collection is involved in a project, an internal review is first conducted to determine if the project falls under one or more 'research' categories that aim to produce "generalizable knowledge" or if it is deemed to be a routine public health activity such as surveillance activities to understand, describe, and contextualize health risks/threats. An exemption is made for projects that are determined to be routine public health activity. Part of the consideration in the determination process is based on how the project was conceived and how the data will be used. This project was conceived by the Sierra Leone Ministry of Health in collaboration with CDC as part of routine mortality surveillance in the post-Ebola epidemic period. The assessment did not aim to produce generalizable knowledge; the aim was to identify context-specific barriers and opportunities in Sierra Leone to improve death reporting as part of routine surveillance following the large Ebola epidemic. However, we should further clarify that regardless of the categorization (i.e. research versus public health surveillance), the same ethical principles for data collection are expected, including proper handling of personal identifying information where applicable. In our assessment, we followed all requisite ethical guidelines for collecting, managing, analyzing, and reporting the data based on CDC guidelines and under the approval of the Sierra Leone Ministry of Health. We had a week-long training for data collectors that comprised a module on ethics, including informed consent. We obtained informed consent from all participants. As mentioned in the previous response, we took careful steps throughout the assessment to anticipate and respond to concerns raised by participants, including potential instances of emotional distress during or after the interview. More detailed information regarding how public health surveillance activities are defined by CDC can be found at website of the US Health and Human Services Department: <https://www.hhs.gov/ohrp/regulations-and-policy/requests-for-comments/draft-guidance-activities-deemed-not-be-research-public-health-surveillance/index.html>

3. Methods:

- the design does not include family members who did report deaths during this period. It would be good to explain why these were not included as they could provide valuable insights; and include in the limitations.

RESPONSE: This qualitative assessment was part of larger efforts to understand death reporting in the aftermath of the end of the Ebola epidemic in Sierra Leone. In the first part of those efforts, we first conducted a telephone survey with individuals who reported to 1-1-7 to describe the deaths captured and motivations for reporting. We then followed up on that phase-1 survey with this phase-2 qualitative assessment to understand the other side of the situation (i.e. why most people did not report). The survey with death reporters has been published and available from: <https://pubmed.ncbi.nlm.nih.gov/32810138>.

- It is unclear why only 66% of respondents were family members of the deceased (as this sampling criteria were household members); who were the other 34%?

RESPONSE: Thank you for identifying this typo. To clarify, 29 out of the 32 interviews were conducted with relatives and only 3 were conducted with non-relative head of households. Though rare, in the context of Sierra Leone, we want to note that it is plausible to have a head of household who is not a blood relative of everyone in the household. We have updated the manuscript accordingly. Also, we have appended the below to Table 1 to provide a full breakdown of respondents' relationship to the deceased person:

Relationship Frequency

Parent 12

Spouse 5

Child 4

Grand parent 3

Non-relative 3

Sibling 3

Other relative 2

Grand Total 32

- It is unclear when these interviews were conducted and they refer to deaths that happen in 2017. Is there a significant lag in time between the death and the interview – what are the implications of this in terms of the perceptions and experiences of the participants?

RESPONSE: The interviews were conducted between August and September 2017. The deaths occurred between April and August 2017 (after the Ebola epidemic / during routine surveillance period). On average, the lag between the death and interview date was between 1 and 3 months. Given that death is a major life event for a family, we do not anticipate that the 1 to 3 months lag biased the results in a substantial way, though there is no way of knowing for sure. The transcripts show that respondents were able to provide very detailed accounts of the events surrounding the death.

4. Results:

- In the results summary – it is unclear what you mean by lack of services linked to reporting – needs to be more clearly aligned with the barrier 1.2 described in page 9.

RESPONSE: We have added an example in parenthesis to say "...lack of services linked to reporting (e.g. provision of ambulance service) ..."

- Were there differences in perceptions and experiences of people living in the 2 districts and the areas within the districts and by gender (as outlined in the sampling section). This has not come across in the results or the discussion.

RESPONSE: No; we added in the results summary that "We did not observe any substantive differences in the thematic findings between districts and areas within districts."

- Perceived inevitability of certain deaths barrier: the quote does not seem to fit in this barrier as it indicates that permission for burials is given by the chief – shows the importance of existing community structures and practices.

RESPONSE: We have dropped the last sentence in the quote as it overlaps with another

sub-theme on localized burial practices. The first part of the quote has been retained because it illustrates perceived inevitability of death (i.e. complications due to stroke).

In addition, more explanation of how people perceived deaths from minor illness, drowning or fire as being inevitable and not being reported.

RESPONSE: We acknowledge the discrepancy and have taken action to try to address it. Those categories of deaths were initially lumped together as deaths that were perceived as not needing to be reported to the 1-1-7 system (i.e. old age, God's will, minor illness, long-term disability, fire, drowning). We later sub-categorized into deaths perceived as inevitable due to the underlying circumstances (e.g. old age, God's will). We now have edited barrier 1.3 accordingly to more accurately focus on perceived inevitability to ensure clarity and cohesion in meaning. We should note that 'minor illnesses' were tied to 'God's will', so we have just retained 'God's will'; for fire and drowning related deaths, participants said that they should be reported to law enforcement so we have therefore removed from barrier 1.3.

5. Discussion:

- this needs to be more aligned with the results emerging from this study.
- if there were differences between the districts these should be discussed.

RESPONSE: We have clarified in the summary of the results that we did not identify any meaningful differences in the themes between and within districts.

- The positive reciprocity section needs to be more related to the findings, and how this can help with mortality reporting going forward.

RESPONSE: We have added the following new paragraph on page 14 to link the results more strongly to the notion of reciprocity:

"Although it took time to establish trust between the Government and communities in achieving high level of death reporting during the Ebola outbreak, 13 35 36 communities had eventually come to expect certain services in return after reporting a death (transportation to the burial ground and laboratory testing of the corpse) and information (communication of laboratory results to the family).²⁸ Although similar services may not be feasible or applicable in the routine mortality surveillance environment, there is an opportunity at the community level to provide aggregated information about the deaths back to the community (e.g. through community leaders and community-based organizations) as a form of reciprocal action to foster dialogue on addressing community level health threats.³⁷ Another key finding from our assessment is that people want help for sick family members who are still alive. Linking the country's expanding fleet of 170 ambulances³⁸ with the 1-1-7 tollfree line could help promote a feeling of reciprocity in addressing other health needs in the community for people experiencing health emergencies."

- There is a suggestion on integrating localised practices for death reporting into routine surveillance systems – can you elaborate how this could be done.

RESPONSE: We have provided an example related to religious and traditional leaders in the conclusion.

"For example, establishing a reporting mechanism through trusted local religious and traditional leaders could help to strengthen reporting levels since these leaders are almost immediately notified of deaths that occur in their communities."

- Including ambulance services as a benefit – is this realistic given the limited resources

within the health system in Sierra Leone?

RESPONSE: Yes, to some extent. With funding from the World Bank and other partners, in 2018, Sierra Leone expanded its ambulance fleet by 170 ambulances that have been allocated to the 16 districts. There is an opportunity to link the 1-1-7 system with the expanded ambulance fleet.

- Mortality surveillance is one aspect that illustrates the level of trust between communities and the government and health system – this should be discussed more, and how this can be improved. Trust in the health system, and reporting system is clearly important but this is not really discussed in the discussion – how do you develop that trust, how is this data used, what are the benefits to the community and how can these be communicated effectively.

RESPONSE: We very much agree. We have incorporated this feedback in numerous places in the revised manuscript based on your previous feedback. For instance, in the second and third paragraphs of the discussion, we highlight how community trust in death reporting evolved over time during the epidemic. In addition, in the fourth paragraph (new) in the discussion we elaborate that: “Although it took time to establish trust between the Government and communities in achieving high level of death reporting during the Ebola outbreak,^{13 35 36} communities had eventually come to expect certain services in return after reporting a death (transportation and laboratory testing of the corpse) and information (communication of laboratory results to the family).²⁸ Although similar services may not be feasible or applicable in the routine mortality surveillance environment, there is an opportunity at the community level to provide aggregated information about the deaths back to the community (e.g. through community leaders and community-based organizations) as a form of reciprocal action to foster dialogue on addressing community level health threats.³⁷ Another key finding from our assessment is that people want help for sick family members who are still alive. Linking the country’s expanding fleet of 170 ambulances³⁸ with the 1-1-7 tollfree line could help promote a feeling of reciprocity in addressing other health needs in the community for people experiencing health emergencies.” Lastly, in the discussion, we provide an example of how trusted religious and traditional leaders can be a conduit for strengthening death reporting through a new stream of reporting to the 1-1-7 system using their existing community networks.